# Preparation and Characterization of Silica Nanoparticles and of Silica-Gentamicin Nanostructured Solution Obtained by Microwave-Assisted Synthesis

**DOI:** 10.3390/ma14082086

**Published:** 2021-04-20

**Authors:** Violeta Purcar, Valentin Rădiţoiu, Cornelia Nichita, Adriana Bălan, Alina Rădiţoiu, Simona Căprărescu, Florentina Monica Raduly, Raluca Manea, Raluca Şomoghi, Cristian-Andi Nicolae, Iuliana Raut, Luiza Jecu

**Affiliations:** 1National Institute for Research and Development in Chemistry and Petrochemistry—ICECHIM, Splaiul Independentei No. 202, 6th District, 060021 Bucharest, Romania; violeta.purcar@icechim.ro (V.P.); coloranti@icechim.ro (A.R.); monica.raduly@icechim.ro (F.M.R.); raluca.manea@icechim.ro (R.M.); r.somoghi@gmail.com (R.Ş.); ca_nicolae@yahoo.com (C.-A.N.); iulia_rt@yahoo.com (I.R.); jecu.luiza@icechim.ro (L.J.); 23Nano-SAE Research Centre, Faculty of Physics, University of Bucharest, 077125 Bucharest-Magurele, Romania; adriana.balan@unibuc.ro; 3National Institute for Chemical-Pharmaceutical Research and Development, Vitan No. 112, 031299 Bucharest, Romania; 4Faculty of Applied Chemistry and Materials Science, University Politehnica of Bucharest, Gh. Polizu No. 1–7, 011061 Bucharest, Romania; simona.caprarescu@upb.ro

**Keywords:** microwave synthesis, silica-gentamicin, morphology, antimicrobial studies

## Abstract

In this research work, silica nanoparticles and silica-gentamicin nanostructured solution were synthesized by using the microwave-assisted synthesis, in basic medium, using two silane precursors (tetraethylorthosilicate and octyltriethoxysilane) and the antibiotic (gentamicin sulfate). The prepared materials were characterized through Fourier transform infrared (FTIR) spectroscopy, TGA analysis, transmission electron microscopy (TEM), and atomic force microscopy (AFM) to investigate the morphology and structure. Antimicrobial studies of the silica-gentamicin nanostructured solution versus silica nanoparticles were performed against *Pseudomonas aeruginosa*, *Staphylococcus aureus*, and *Escherichia coli*. FTIR spectra showed that the gentamicin has been loaded to the silica nanoparticles. AFM analysis showed that the morphology of the silica-gentamicin nanostructured solution has changed, and agglomerations of particles are present at the surface. Antimicrobial testing, performed using the diffusion method through spot inoculation, indicates that the silica-gentamicin nanostructured solution exhibited activity against the resistant strain. The obtained silica-gentamicin solution can be used as biochemical agent for the prevention and treatment of microorganisms which are deposited on different surfaces (e.g., glass, plastic, ceramic).

## 1. Introduction

In recent years, silica nanoparticles (SiO_2_ NPs) have received much attention because they are a promising candidate for drug delivery, gene therapy, the detection of biomolecules, photodynamic therapy, and bioimaging [1,2]. It has been demonstrated that the SiO_2_ NPs can be used as a system to deliver drugs such as antibiotics due to high thermal, chemical and colloidal stability, high surface area, and good biocompatibility [3]. SiO_2_ NPs have attracted the attention of researchers because they can easily modify through established organosilane chemistry, which allowed the incorporation of functional groups [4]. Pedraza et al. [5] showed that the mesoporous silica nanoparticles (MSNs), loaded with an antimicrobial agent (levofloxacin) and functionalized with aminopropyltrimethoxysilane (DAMO, targeting agent), can be used as antibiotic nanocarrier able to penetrate bacterial biofilm. Huh and Kwon demonstrated that the immobilization of drugs in nanoparticles can reduce adverse side effects [6]. Silica nanoparticles functionalized with antibiotic can be suitable for use as a biochemical agent [7,8]. Gounani et al. [9] indicated that the antibiotic molecules-loaded nanoparticles enhanced antibacterial activity compared to free antibiotic (polymyxin B) against different Gram-negative bacteria. It was shown that the antibacterial agents developed via grafting or encapsulation processes can locally destroy bacteria, without being toxic to the surrounding cells and tissues [10]. Gentamicin is a potent broad-spectrum antibiotic that is effective against Gram-positive and Gram-negative bacteria, belonging to the aminoglycoside class. Gentamicin is employed to treat various infections such as orthopedic and ocular [11]. Mosselhy et al. demonstrated that the silica-gentamicin delivery systems can be used as a biocompatible drug delivery system in combating planktonic methicillin-resistant *Staphylococcus aureus* cells and eradicating *E. coli* biofilms [12]. Silica nanoparticles loaded with gentamicin could be prepared by using different synthesis routes, resulting in various levels of drug loading and different release kinetics. Barbé et al. [13] prepared the silica-gentamicin nanohybrids and native SiO_2_ NPs through a single step sol-gel procedure, at ambient temperature. They demonstrated that the sol-gel technology can provides enhanced flexibility in both the encapsulation and the release of drug molecules. Corrêa et al. [14] investigated the encapsulation of gentamicin by acid-catalyzed gelation, base-catalyzed gelation, and base-catalyzed precipitation routes. The antimicrobial effects of encapsulated gentamicin samples were tested against a series of Gram-positive and Gram-negative bacterial strains. It has been shown that the sol-gel route affects the elemental, structural, textural, and morphological characteristics of the prepared materials. Thaher Al [15] prepared silica nanoparticles with gentamicin through different routes (layer-by-layer, entrapment, adsorption). The studies indicated that the coating technique layer-by-layer achieved better loading and control for drug release in comparison with the entrapment and adsorption routes. It was demonstrated that the gentamicin released from silica nanoparticles can be controlled in different conditions (two buffer media, different polyelectrolytes). Tariq et al. [16] show that the modified mesoporous silica nanoparticles preloaded with gentamicin can be synthesized through the emulsion electrospinning method. They related that the electrospun nanofibers of the functionalized silica preloaded with gentamicin showed a high encapsulation efficiency and controllable release profiles. The studies indicated that almost half of the gentamicin (54%) was released for two weeks. Chi-Jen Shih et al. [17] evaluated the gentamicin encapsulated powders of hierarchically meso-macroporous silica-based calcium phosphate glass. The studies indicated the fact that the increasing the initial concentration of gentamicin reduced the encapsulation efficiency. The maximum value of encapsulated gentamicin (39.9 mg/g) was obtained when an initial concentration of 50 mg/mL was used. Nampi et al. [18] studied the elution of different quantities of the gentamicin incorporated into a silica matrix, which was obtained by the sol-gel route. Based on the release properties, they reported that a silica matrix can be used as a substrate for drug release. Tamanna et al. [19] demonstrated that the thin-film coatings embedded with gentamicin-loaded mesoporous silica nanoparticles provide antibacterial and anti-biofilm activity over a prolonged period. Zhou et al. [20] demonstrated that the composition of mesoporous bioactive glasses had an influence on gentamicin sulfate loading and release rate.

The microwave-assisted technique has been applied for the synthesis of materials such as metal oxides [21], metallic nanomaterials [22], metal organic frameworks [23], or polymers [24]. It was proved that the microwave approach is extremely fast, providing good to excellent yields (82–95%) when compared with the conventional method (45–68%) [25]. The hydrothermal synthesis of materials using microwave can reduce the synthesis time, can reduce significantly particle size, and can reduce energy consumption in comparison with the conventional convection heating method [26] and conventional autoclave heating [27]. The conventional systems lead to an uneven temperature distribution due to poor heat transfer into the bulk of the obtained material [27]. 

The microwave-assisted technique offers many benefits, such as rapid volumetric heating, high reaction rates, spontaneous nucleation events, reproducibility, and size and shape control by tuning reaction parameters. Microwaves induce an intense electric field that generates dynamic dipole moments in the molecules in the reaction medium. These molecules are polarized and begin to oscillate and generate heat. Based on these considerations, the advantage of microwave-assisted synthesis is that the energy is completely transformed into heat. The heating is direct, almost instantaneous, and does not require heat transfer, as in the case of synthesis with classical heating and the process takes place in a very short time. Laurenti et al. [28] demonstrated that the adsorption of gentamicin sulfate (GS) within the zinc oxide (ZnO) matrix was successfully occurred. The results showed that the mesoporous ZnO matrix presented different behaviors and drug-loading efficiencies, depending on the GS loading solution. Kamarudin et al. [29] showed that the mesoporous silica nanoparticles, prepared by using microwave synthesis, exhibited the ibuprofen adsorption. It was observed that the mesoporous structure of silica nanoparticles can be modified as function of the used microwave power (from 100 to 450 W). The highest adsorption rate of 98.3% was obtained for mesoporous silica nanoparticles at 450 W after 7 h. In our previous studies, we showed tha.t the silica nanoparticles can be successfully obtained through microwave-assisted synthesis (maximum power to 200 W, temperature of 50 °C), using silane precursors at different molar ratios (5:1 and 10:1, respectively). The results revealed that the obtained silica nanoparticles present different average particle sizes, ranging from 138.5 to 697.5 nm. The solvent, silica precursors, catalyst, and microwave irradiation time were properly selected to obtain modified silica nanoparticles with hydrophobic property [30]. Based on research and our knowledge, microwave-assisted synthesis of silica nanoparticles encapsulated with gentamicin was not previously been reported in the literature. In this study, we report the preparation of silica nanoparticles (SiO_2_ NPs) and of silica-gentamicin nanostructured solution through microwave-assisted synthesis in basic medium. For this purpose, two silane precursors (tetraethylorthosilicate (TEOS) and octyltriethoxysilane (OTES)) were used to synthesize the silica nanoparticles. Silica nanoparticles prepared in this way will act as an efficient carrier of biomolecules. Gentamicin sulfate (bactericidal aminoglycoside antibiotic) was selected as a drug molecule because of its use in the treatment of bacterial infections. This antibiotic was encapsulated within plain SiO_2_ NPs in order to obtain silica-gentamicin nanostructured solution. Microwave-assisted synthesis was chosen to realize these materials because it is inexpensive, fast, and totally non-polluting method. In this paper, it was shown that under microwave radiation, the silica nanoparticles and silica-gentamicin nanostructured solution were obtained in short reaction time (40 min), compared with conventional methods. These obtained materials were subjected to physicochemical and morphological characterization followed by evaluation for their antimicrobial activity on Gram-positive (*Staphylococcus aureus*) and Gram-negative (*Pseudomonas aeruginosa* and *Escherichia coli*) bacteria. The results showed that the silica-gentamicin nanostructured solution possessed antimicrobial activity against all the bacterial strains. These final materials can be used for various applications in nanotechnology.

## 2. Materials and Methods

### 2.1. Materials

Gentamicin sulfate (solubilized in water, 3%, GS, Sigma-Aldrich, St. Louis, MO, USA) was used as an antibiotic. Tetraethylorthosilicate (TEOS, 98%, Aldrich, St. Louis, MO, USA) was used as the silica particle precursor). Octyltriethoxysilane (OTES, 97%, Fluka, Philadelphia, PA, USA) was used as the modifying agent. Isopropyl alcohol (iPrOH, 99.9%, Chimreactiv S.R.L., Bucharest, Romania) was used as the solvent. Aqueous solution of ammonia (NH_4_OH, 25%, Sigma-Aldrich, Merck KGaA, Darmstadt, Germany) was used as the catalyst. The chemicals were used as received. The glass substrate was purchased from FabTech (Bucharest, Romania) and was chosen in order to investigate the topography of films obtained by deposition of final materials.

Three fungal strains, *Staphylococcus aureus*, *Pseudomonas aeruginosa*, and *Escherichia coli*), purchased from German Collection of Microorganisms and Cell Cultures (DSMZ) (Braunschweig, Germany), were used in the experiments.

### 2.2. Preparation of Silica Nanoparticles and of Silica-Gentamicin Nanostructured Solution by Microwave-Assisted Synthesis

Silica nanoparticles (SiO_2_ NPs), with and without gentamicin sulfate (GS), were synthesized by a base catalyzed process using a microwave-assisted system.

For the synthesis of silica nanoparticles (SiO_2_ NPs) without gentamicin sulfate (GS), in the first step, isopropyl alcohol (iPrOH, 19.5 mL), distilled water (1.8 mL), and aqueous solution of ammonia (NH_4_OH 25%, 1.02 × 10^−2^ moles) were stirred together in the microwave-assisted system for 10 min, at 50 °C. As the second step, an alcoholic solution of tetraethylorthosilicate (TEOS, 1.34 × 10^−3^ moles) in isopropyl alcohol (1.56 mL) was rapidly added to the above mixture and stirring at 50 °C was continued until a slight opalescence appeared (approximately 10 min). Then, octyltriethoxysilane (OTES, 1.34 × 10^−3^ moles) was added dropwise to the reaction mixture. Stirring and temperature (50 °C) were maintained for another 20 min.

The silica-gentamicin nanostructured solution was prepared according to the same protocol of silica nanoparticles except for the addition of amount of gentamicin sulfate (1 mL) in the first step of synthesis. The final nanostructured solution was homogeneous and opalescence (see Scheme 1).

Materials used for synthesis of silica nanoparticles (SiO_2_ NPs) and of silica-gentamicin nanostructured solution are presented in Table 1. The resulted samples were characterized as dispersions, as powders (obtained after solvent evaporation), and as films (deposited onto clean glass substrates). The glass substrates were cleaned by regular soap solution in an ultrasonic water bath for 30 min followed by washing with distilled water and ethanol. This washing process was repeated three times. Finally, the glass substrates were vacuum-dried and stored in desiccator for 24 h in order to ensure uniform wetting.

All final samples (powders and films) were dried and kept (overnight) at room temperature (25 °C) and then characterized in order to investigate the structural and morphological properties. In addition, samples were characterized as dispersions in order to observe the antimicrobial activity.

### 2.3. Characterization of Silica Nanoparticles and of Silica-Gentamicin Nanostructured Solution

Silica nanoparticles (SiO_2_ NPs), with and without gentamicin sulfate (GS) were synthetized in a microwave-assisted system (Microwave Labstation for Synthesis, Milestone, Sorisole, Italy), with the maximum power set to 200 W, frequency of 2.45 GHz, temperature of 50 °C, time of 40 min.

The structure’s modifications of the obtained materials were carried out on a Jasco FTIR 6300 instrument (JASCO Int. Co., Ltd., Tokyo, Japan), with an integrating sphere, detector wide MCT (Pike Technologies Inc., Fitchburg, MA, USA). The samples were ground with KBr powder and pressed to form a disc for FTIR scanning. Data were collected in cooled liquid nitrogen, in a wavenumber range of 500–5000 cm^−1^ (160 scans at a resolution of 4 cm^−1^).

Thermal analysis of the final samples (obtained as powders, 5–10 mg) was performed in nitrogen atmosphere (heating rate of 10 °C/min), 40–700 °C range, using a TA TGA Q5000 IR instrument (TA Instruments, New Castle, DE, USA).

Morphology and shape of silica nanoparticles and of silica-gentamicin nanostructured solution (obtained as dispersions) were studied by Transmission Electron Microscopy (TEM, FEI Company, Phillips, The Netherlands).

Morphology and topography of resulted materials (obtained as films by deposition of final solutions onto glass substrates) were recorded through Atomic Force Microscopy (NTEGRA PRIMA Platform (NT-MDT), Moscow, Russia) in semi-contact mode (scanning area range 2.5 × 2.5 µm^2^), using an NSG01 cantilever (resonance frequency: 87–230 kHz, force constant: 1.45–15.1 N/m). In addition, the surface morphology of these films was observed using a digital optical microscope (LCD Digital Microscope II, full color 3.5″ TFT LCD screen, Celestron, Poland).

The antibacterial activity was tested by the diffusion method through spot inoculation. The tests were carried out with bacterial strains, as *Pseudomonas aeruginosa*, *Staphylococcus aureus,* and *Escherichia coli*. The strains were inoculated into canvas on solid Muller–Hinton [31] medium distributed in Petri dishes. The composition of solid Mueller–Hinton medium (g·L^−1^) was 2, meat extract; 1.5, starch; 17.5 casein hydrolysate; 17, agar; 1000 mL distilled water (final pH = 7.2–7.4). As inoculum, it was used a suspension in the sterile physiological water made from a fresh culture of 18–24 h (4–5 isolated colonies) developed on a solid medium, with a density of 1 × 10^8^ CFU/mL adjusted nephelometrically (McFarland standard 0.5 = 1.5 × 10^8^ UFC/mL). The tested compounds were added in spot, in a volume of 10 µL. Petri dishes were incubated at 36 ± 1 °C for 24 h. After 24 h, the antimicrobial activity was evaluated by measuring the diameter of the clear area (halo) appearing around the inoculation area (spot). The tests were performed in duplicate.

## 3. Results

Silica nanoparticles (SiO_2_ NPs) and silica-gentamicin nanostructured solution, prepared by the base-catalyzed route, using a microwave-assisted system, were characterized by using different series of instrumental characteristic techniques to investigate the structural and morphological properties, and antimicrobial activity, respectively.

### 3.1. FTIR Spectroscopy

The FTIR spectra of gentamicin sulfate (GS), SiO_2_ NPs (sample G0), and silica-gentamicin nanostructured solution (sample G1) are depicted in Figure 1. The samples were characterized as powders.

In spectrum of gentamicin sulfate (GS), two peaks at 1530 and ≈1630 cm^−1^ were detected, and it can be assigned to cyclic amines or amine groups bonded to an aliphatic ring [32]. The free gentamicin showed a band at 603 cm^−1^ that is considered a major band for gentamicin. Peak at around 3400–3600 cm^−1^ corresponds to N–H stretching mode of H-bonded amide group. Spectra of SiO_2_ NPs (sample G0) and of silica-gentamicin nanostructured solution (sample G1) showed peaks at 2925 cm^−1^ and 2850 cm^−1^, respectively, attributed to the C–H stretching modes (–CH_2_ and –CH_3_, symmetrical and asymmetrical stretching). The peak seen approximately at 795 cm^−1^ corresponds to the symmetric stretching vibration of the Si–O–Si network of silica [33]. The spectrum of sample G1 showed peaks at 1638, 1555, and 1379 cm^−1^ that indicate the incorporation of gentamicin onto the silica matrix. The characteristic peaks at 1122 and 1045 cm^−1^ were assigned to asymmetric Si–O–Si stretching vibrations [18,34].

### 3.2. TGA Analysis

The thermogravimetric analysis (TGA) results of samples obtained as powders (gentamicin sulfate (GS), SiO_2_ NPs (sample G0) and silica-gentamicin nanostructured solution (sample G1)) are shown in Figure 2.

To determine several significant effects, the weight loss (Wt. loss) and maximum decomposition temperature (*T*_max_) were studied over three temperature intervals: 40–156 °C, 156–262 °C, and 262–700 °C (see Table 2). The TGA result of gentamicin sulfate is depicted in Figure 2a. One can note that the temperature of ≈240 °C can be considered as the beginning of gentamicin decomposition [35]. The initial weight loss of 2.87% and 2.36% during heating of the SiO_2_ NPs (sample G0) and of the silica-gentamicin nanostructured solution (sample G1) up to 100 °C can be induced by the removal of the absorbed and residual water. The data reveal a total mass loss of 29.23% and 48.63% at ≈500 °C for the SiO_2_ NPs and for sample G1, respectively, which might be due to the loss of alkyl groups.

### 3.3. Microscopic Studies

The size and morphology of the SiO_2_ NPs (sample G0) and silica-gentamicin nanostructured solution (sample G1) were visualized by transmission electron microscopy (TEM), as shown in Figure 3. The samples were characterized as dispersions. TEM image of the sample G0 showed that the silica nanoparticles are quasi-spherical with smooth surfaces. Analyzing the TEM image of sample G1, it can be observed that the silica-gentamicin nanostructured solution increases in the size over that of the SiO_2_ NPs (sample G0). This fact demonstrated the successful loading of gentamicin onto the surface of SiO_2_ NPs and the encapsulation of some gentamicin within the SiO_2_ network. This result is in good agreement with the literature [36], demonstrating that the silica shell deposition was favored by the positively charged of gentamicin on the silica particle surface. Kamarudin et al. [29] showed that the variation in microwave power led to modification in the material’s structure. The studies indicated that the complete adsorption of ibuprofen that was obtained using mesoporous silica nanoparticles prepared at microwave power of 450 W was possible due to the higher surface area and fewer irregularities in the morphology. Van de Belt et al. [37] demonstrated that the relationship between the surface roughness of gentamicin-loaded carriers and the antibiotic release is very important because the rougher surfaces establish a larger area for antibiotic release from the surfaces of carriers for infection prevention. Inada et al. showed that spherical mesoporous SiO_2_ particles can be synthesized by the sol-gel method using water/oil emulsion under microwave irradiation but using acetylacetone as capping agent [38].

These results indicate that the rapid and selective heating by microwave irradiation (temperature of 50 °C, microwave power of 200 W) is effective to make the silica nanoparticles and the silica-gentamicin nanostructured solution without the use of expensive agents.

The surface morphology of the films (obtained by deposition of the silica nanoparticles (sample G0) and of the silica-gentamicin nanostructured solution (sample G1) on glass substrates) was investigated by AFM analysis (Figure 4). The 3D AFM images from Figure 4 can be understood as topographic maps of the samples.

Analyzing Figure 4, it can be observed that the size of the silica nanoparticles (sample G0) was approximately 400 nm. One can note that the morphology of the silica-gentamicin nanostructured solution (sample G1) has changed, and agglomerations of particles are present at the surface, confirmed also by the optical microscope images.

The morphological examination of these films was also performed using a digital optical microscope (magnification 100×) (Figure 5).

As illustrated in Figure 5, the silica-gentamicin nanostructured solution (sample G1) shows many differently shaped particles that are distributed over the entire glass substrate. In addition, agglomerations of particles can be observed, which is possibly due the interaction and interconnection between gentamicin and silica nanoparticles (Si–O–Si network with methyl- or ethyl-organic groups) [39]. The interconnection (presence of silanol group at the surface of silica nanoparticles linked with the –COOH or –NH_2_ group of gentamicin) could be a surface phenomenon via weak interactions of van der Waals force or hydrogen bonding [40]. This can lead to the filling and increasing of pores and dense microstructure after adding gentamicin. Ballarre et al. [39] reported that the agglomerations of particles can promote the increase of unevenness. Moreover, the agglomeration leads to particle sedimentation and provides instability. The adding of the gentamicin molecules in the obtained material led to the increasing of pores, which is possibly due to the increase of amount of gentamicin. In addition, the microstructures of silica-gentamicin nanostructured solution become dense due to the insertion of gentamicin in the pores as well as due to its adherence on the glass substrates [39,40]. It was demonstrated that the adsorption of gentamicin molecules into mesoporous channels are based on the adsorptive properties of silanol groups while the surface area of silica-gentamicin nano-solution is correlated with the amount of gentamicin added [41].

### 3.4. Antimicrobial Studies

Antimicrobial studies of the silica-gentamicin nanostructured solution (sample G1) versus SiO_2_ NPs (sample G0) were performed against *Pseudomonas aeruginosa*, *Staphylococcus aureus,* and *Escherichia coli*. Antibiograms were carried out using the diffusion method through spot inoculation (Figure 6). The inhibitory effect of gentamicin on bacteria was determined by measuring the diameter of the inhibition zone (see Table 3). The inhibition zone diameters were measured after 24 h. As analyzed in Figure 6, SiO_2_ NPs (sample G0) did not exhibit antibacterial activity against bacterial strains. The diffusion method through spot inoculation results showed that free gentamicin has a marked antimicrobial effect against all bacteria. The antimicrobial efficacy of the silica-gentamicin nanostructured solution is less than that of the free gentamicin. The diameter of the inhibition zone for silica-gentamicin solution was of ≈18 mm. These results are in good agreement with the literature showing that the gentamicin influences the formation of SiO_2_ NPs through attractive electrostatic interactions, leading to larger particles and higher loadings [42]. In addition, this fact can be due to the aforementioned release profile of gentamicin from the silica-gentamicin nanostructured solution. Corrêa et al. [14] studied the antibacterial activity of silica-encapsulated gentamicin. They observed that the encapsulation method appeared to affect the activity of the *S. aureus*, in which the inhibition diameter increases from 14.93 mm (not-encapsulated) to 21.80 mm (1000 mg encapsulated gentamicin), which was possibly due to the surface and textural characteristics of the obtained material.

Our data demonstrate that the silica-gentamicin solution fits as an ideal antimicrobial agent with the potential to minimize the healthcare and environmental-related problems.

## 4. Conclusions

In conclusion, we have demonstrated that the silica nanoparticles and the silica-gentamicin nanostructured solution can be prepared by the microwave-assisted synthesis in basic medium. The FTIR results indicated that the gentamicin has been loaded to the silica nanoparticles (SiO_2_ NPs). The TGA data demonstrated a total mass loss of 29.23% and 48.63% at ≈500 °C for the SiO_2_ NPs (G0) and silica-gentamicin nanostructured solution (G1), respectively, which can be due to the loss of alkyl groups. TEM images showed that the silica-gentamicin nanostructured solution presents the different morphology than that of SiO_2_ NPs and suggest the loading of the antibiotic to the SiO_2_ network. Antimicrobial study revealed that the silica-gentamicin nanostructured solution exhibited activity against *Pseudomonas aeruginosa*, *Stapylococcus aureus*, and *Escherichia coli* strains. The diffusion method through spot inoculation revealed that the diameter of the inhibition zones was of about 18 mm for sample G1. SiO_2_ NPs (G0) did not exhibit antibacterial activity, and no zones of inhibition were produced against any of the tested bacterial species. The microwave-assisted synthesis provides very facile and rapid functionalization of silica nanoparticles with antibiotic. The silica-gentamicin solution obtained by microwave-assisted synthesis can be used as biochemical agent for the prevention and treatment of microorganisms that are deposited on different surfaces (e.g., glass, plastic, ceramic).

## Data Availability

The data are not publicly available due to their containing information that could compromise the privacy of research participants.

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
