# Peer review of "Preparation and Characterization of Silica Nanoparticles and of Silica-Gentamicin Nanostructured Solution Obtained by Microwave-Assisted Synthesis"

_materials, 2021, doi:10.3390/ma14082086_

Round 1

Reviewer 1 Report

The paper itself is written adequately with sufficient data and discussion. However, the biggest flaw of the paper is in the concept of the paper itself, as it has a glaring inconsistency between the main, proncipal novelty of the paper, which is the use of microwave synthesis, and the results presented, which contain no exploration of the microwave parameters and synthesis routes. The premise of the paper is microwave synthesis has not been done for silica/gentamicin, as 'conventional' NPs has been done and can easily be searched online for relevant publications. However, the authors do not explore the microwave synthesis itself, and as such the reader has no idea whether microwave synthesis has any benefit compared to conventional synthesis. Other papers claim very similar result when it comes to material size and properties, and in this paper there is also no positive control (gentamicin without NP) so the effect of the silica itself is not fully explored. 

I would recommend that the authors add more discussion in their results on whether their microwave synthesis has benefits in terms of time, temperature, or other aspects which may affect commercial pharmaceutical production, and add comparisons between their synthetic method and the conventional ones present in the literature in order to distinguish this paper from previously published materials. As it stands, in the results and discussion there is only one mention of microwave in the very start without any further exploration.

Author Response

The paper itself is written adequately with sufficient data and discussion. However, the biggest flaw of the paper is in the concept of the paper itself, as it has a glaring inconsistency between the main, proncipal novelty of the paper, which is the use of microwave synthesis, and the results presented, which contain no exploration of the microwave parameters and synthesis routes. The premise of the paper is microwave synthesis has not been done for silica/gentamicin, as 'conventional' NPs has been done and can easily be searched online for relevant publications. However, the authors do not explore the microwave synthesis itself, and as such the reader has no idea whether microwave synthesis has any benefit compared to conventional synthesis. Other papers claim very similar result when it comes to material size and properties, and in this paper there is also no positive control (gentamicin without NP) so the effect of the silica itself is not fully explored. 

I would recommend that the authors add more discussion in their results on whether their microwave synthesis has benefits in terms of time, temperature, or other aspects which may affect commercial pharmaceutical production, and add comparisons between their synthetic method and the conventional ones present in the literature in order to distinguish this paper from previously published materials. As it stands, in the results and discussion there is only one mention of microwave in the very start without any further exploration.

Response

Thank you for the constructive comments.

New information’s about the idea whether microwave synthesis has any benefit compared to conventional synthesis were added:

  • page 3: “The hydrothermal synthesis of materials using microwave can reduce the synthesis time, can reduce significantly particle size, can reduce energy consuming in comparison with the conventional convection heating method [26] and conventional autoclave heating [27]. The conventional systems leads to an uneven temperature distribution due to poor heat transfer into the bulk of the obtained material [27].”
  • page 3: “Microwaves induce an intense electric field that generates dynamic dipole moments in the molecules in the reaction medium. These molecules are polarized and begin to oscillate and generate heat. Based on these considerations, the advantage of microwave-assisted synthesis is that the energy is completely transformed into heat. The heating is direct almost instantaneous does not require heat transfer, as in the case of synthesis with classical heating and the process takes place in a very short time.”

In this paper was shown that under microwave radiation the silica nanoparticles and silica-gentamicin nanostructured solution were obtained in short reaction time (40 minutes), compared with conventional methods.

The comparisons between our synthetic method and the conventional ones present in the literature in order to distinguish this paper from previously published materials.

  • page 9: “Kamarudin et al. [29] shown that the variation in microwave power led to modification in the material’s structure. The studies indicated that the complete adsorption of ibuprofen was obtained using mesoporous silica nanoparticles prepared at microwave power of 450 W possible due to the higher surface area and fewer irregularities in the morphology.
  • page 9: “Inada et al. shown that spherical mesoporous SiO2 particles can be syntjetized by sol–gel method using water/oil emulsion under microwave irradiation, but using acetylacetone as capping agent [35].”
  • These results indicate that the rapid and selective heating by microwave irradiation (temperature of 50 °C, microwave power of 200 W) is effective to make silica nanoparticles and silica-gentamicin nanostructured solution without the use of expensive agents.

Reviewer 2 Report

The work entitled "Preparation and characterization of silica-gentamicin nanohybrids obtained by microwave synthesis" by Violeta Purcar, Valentin Rădiţoiu, Cornelia Nichita, Adriana Bălan, Alina Rădiţoiu, Simona Căprărescu, Florentina Monica Raduly, Raluca Manea, Raluca Şomoghi, Cristian-Andi Nicolae, Mariana Călin, Luiza Jecu present an interesting work on functionalization of silica NPs with gentamicin. Generally, the work contains coherent and reliable results however general concept and research hypothesis is missing together with some details. Also the scientific impact is low for this reason, but additionally a deeper discussion is recommended.

Here are my comments.

Research hypothesis and explanation why silica nanoparticles are beneficial for the functionalization with antibiotics is missing both in the introduction and abstract.

I would rather discourage using old naming, even if it is found on commercial products like Ammonium hydroxide solution NH4OH. Aqueous solution of ammonia would much better refer to general chemical properties of the substance.

Providing speed of rotation (400 rpm) is useless, since the stirring effect is dependent on many factors like the shape of vessel, stir bar type, etc.

Information “The glass substrates were cleaned with soap” is very general and details are missing.

TGA analysis revealed mass loss which was associated with alkyl group decomposition. However the main question is what is the gentamicin load onto the nanoparticles. At least a rough assessment is necessary.

The final conclusion is irrelevant: “These obtained materials can be used as antimicrobial agents to 292 prevent the infections of the surfaces with bacteria/microbes.” It is a general conclusion not supported by the results.

There is a general issue missing in this work: What is the hypothesis of the research? Why silica NPs have been chosen to support the antibiotic? Why they are expected to be better than any other materials? What is the purpose to anchoring the antibiotics on the NPs? What are the perspective applications or any benefits from such an approach?

I do not recommend this for publication unless these points will be addressed. Therefore, my recommendation is a major revision.

Author Response

The work entitled "Preparation and characterization of silica-gentamicin nanohybrids obtained by microwave synthesis" by Violeta Purcar, Valentin Rădiţoiu, Cornelia Nichita, Adriana Bălan, Alina Rădiţoiu, Simona Căprărescu, Florentina Monica Raduly, Raluca Manea, Raluca Şomoghi, Cristian-Andi Nicolae, Mariana Călin, Luiza Jecu present an interesting work on functionalization of silica NPs with gentamicin. Generally, the work contains coherent and reliable results however general concept and research hypothesis is missing together with some details. Also the scientific impact is low for this reason, but additionally a deeper discussion is recommended.

Here are my comments.

Research hypothesis and explanation why silica nanoparticles are beneficial for the functionalization with antibiotics is missing both in the introduction and abstract.

Response

Research hypothesis and explanation why silica nanoparticles are beneficial for the functionalization with antibiotics was added in the manuscript:

- Abstract, page 1: “The obtained silica-gentamicin solution can be used as biochemical agent for the prevention and treatment of infections due to their effective biocidal activity and environmentally friendly performance.”

- Introduction, page 3: “The hydrothermal synthesis of materials using microwave can reduce the synthesis time, can reduce significantly particle size, can reduce energy consuming in comparison with the conventional convection heating method [26] and conventional autoclave heating [27]. The conventional systems leads to an uneven temperature distribution due to poor heat transfer into the bulk of the obtained material [27].”

I would rather discourage using old naming, even if it is found on commercial products like Ammonium hydroxide solution NH4OH. Aqueous solution of ammonia would much better refer to general chemical properties of the substance.

Response

  • sections 2.1. and 2.2., page 4: “Ammonium hydroxide solution NH4OH” was replaced with “Aqueous solution of ammonia”.

Providing speed of rotation (400 rpm) is useless, since the stirring effect is dependent on many factors like the shape of vessel, stir bar type, etc.

Response

The parameters for microwave-assisted synthesis were added:

  • section 2.3, page 5 “frequency of 2.45GHz, temperature of 50 °C, time of 40 minutes.”

Information “The glass substrates were cleaned with soap” is very general and details are missing.

Response

The new details were added:

  • section 2.2, page 4: “The glass substrates were cleaned by regular soap solution in an ultrasonic water bath for 30 min. followed by washing with distilled water and ethanol. This washing process was repeated three times. Finally the glass substrates were vacuum-dried and stored in desiccator for 24 h in order to ensure uniform wetting.”

TGA analysis revealed mass loss which was associated with alkyl group decomposition. However the main question is what is the gentamicin load onto the nanoparticles. At least a rough assessment is necessary.

Response

Thank you very much for your suggestion!

TGA analysis was replaced with the new figure. New information’s were added.

The final conclusion is irrelevant: “These obtained materials can be used as antimicrobial agents to 292 prevent the infections of the surfaces with bacteria/microbes.” It is a general conclusion not supported by the results.

Response

The new final conclusion was added: “This silica-gentamicin solution obtained by microwave-assisted synthesis can be used as biochemical agent for the prevention and treatment of infections due to it effective biocidal activity and environmentally friendly performance.”

There is a general issue missing in this work: What is the hypothesis of the research? Why silica NPs have been chosen to support the antibiotic? Why they are expected to be better than any other materials? What is the purpose to anchoring the antibiotics on the NPs? What are the perspective applications or any benefits from such an approach?

Response

New information’s were added:

  • page 3: “The hydrothermal synthesis of materials using microwave can reduce the synthesis time, can reduce significantly particle size, can reduce energy consuming in comparison with the conventional convection heating method [26] and conventional autoclave heating [27]. The conventional systems leads to an uneven temperature distribution due to poor heat transfer into the bulk of the obtained material [27].”
  • page 3: “Microwaves induce an intense electric field that generates dynamic dipole moments in the molecules in the reaction medium. These molecules are polarized and begin to oscillate and generate heat. Based on these considerations, the advantage of microwave-assisted synthesis is that the energy is completely transformed into heat. The heating is direct almost instantaneous does not require heat transfer, as in the case of synthesis with classical heating and the process takes place in a very short time.”
  • In this paper was shown that under microwave radiation the silica nanoparticles and silica-gentamicin nanostructured solution were obtained in short reaction time (40 minutes), compared with conventional methods.
  • Silica nanoparticles prepared in this way will act as an efficient carrier of biomolecules.
  • These final materials can be used for various applications in nano medicine.
  • These results indicate that the rapid and selective heating by microwave irradiation (temperature of 50 °C, microwave power of 200 W) is effective to make silica nanoparticles and silica-gentamicin nanostructured solution without the use of expensive agents.

Reviewer 3 Report

In the present article, Purcar et al. report the synthesis, characterization and antimicrobial activity of gentamicin-loaded silica nanoparticles. The manuscript is concise, the microscopy studies are impressive, however the work lacks in terms of physicochemical characterizations.

  1. Abstract

- line 20: continuous microwave process. In general a continuous process refers to a process where there is a continuous stream of reactants and products in contrast to a batch process. The authors should use a different term.

  1. Introduction

- lines 87-89. The phrase “controlled power quickly to the reaction” should be rewritten. The phrase “uniform heating by mechanisms” is incomplete.

- lines 92-94. If the authors want to include this reference, they should elaborate slightly more on it.

  1. Section 2.2

- Quantities of TEOS, OTES and NH4OH, which I guess is the catalyst in the sol-gel process performed by the authors, should be expressed in moles (or mmoles).

- How can the authors explain the formation and isolation of nanoparticles? In general, tedious processes are implemented to obtain nanoparticles with a narrow size distribution, while in this manuscript is seems like the authors just added the silane precursors and particles are formed.

- Were the particles washed somehow to removed unreacted silane precursors?

- Lines 137-138: is gentamicin sulfate a liquid (in general salts are solids)? If it is a solid, authors should specify the solvent where GS was dissolved and the concentration of the solution. Why were these quantities used? Did the authors try other ones?

- Table 1. The theoretical content of gentamicin in the particles should be calculated and added to the table. The experimental content should be measured and added as well.

  1. Section 2.3

- If the authors have used a microwave probe, it should be specified somewhere.

- How many scans where taken for the IR spectra? The authors should also add how the spectra recorded (i.e., KBr tablets, nujol, something else).

  1. Section 3.1

- IR spectra are typically recorded up to 4000 cm-1 and interesting absorption bands can be observed in that region, why do the authors stop at 3200 cm-1?

- The peaks at 1525 and 1630 cm-1 are too weak to be used for characterization. Peak at 1525 cm-1 is not even indicated in the spectrum. A lot of peaks of similar intensity are observed in that region. Amine absorption should be observed around 3400-3600 cm-1 and, often, a different pattern is observed for primary and secondary amine groups. Additionally, to the best of my knowledge, gentamicin does not bear any aromatic rings. In any case, a structure of gentamicin must be added.

- In the 2850-2950 region, attributed to C-H stretching modes. Similar absorption bands should be observed in the GS spectrum, but there aren’t any. Why? Why is the intensity of the absorption band of G1 so much more intense compared to G0?

- According to the authors, the peak at 1620 cm-1 represents both loaded GS and adsorbed water. This peak is much too small to be used for such an assertion. Water in general is not observed in this region of the IR spectrum.

- To which specifical chemical bonds of nanosilica is the peak at 1460 cm-1 attributed to? Why is a similar peak observed on the spectrum of GS?

- To which chemical group is the band at 960 cm-1 attributed to? This peak is present in all three spectra.

- The small peak at 667 cm-1 could indeed be attributed to gentamicin and indicated the successful presence of GS in the silica particles, but the authors have to specify to which chemical group it belongs if they want to use it for their analysis.

- Finally, the authors concentrate on very weak peaks, while they do not comment at all on more intense peaks such as the peak around 1700 cm-1 in G1, the peak at ca. 1108 cm-1 in all spectra. These peaks must absolutely be attributed. Additionally, the peaks at 739 and 610 cm-1 should also be discussed.

  1. Section 3.2

- The TGA analysis of GS should be added in Figure 2.

- Why is the degradation of GS observed in G1 and not in G2, which, according to the authors, contains more GS?

- Why is the mass loss more important in G1 compared to G0? G1 should have a higher organic content since it also contains GS.

- Is the residual mass consistent with the amount of OTES that is used for the preparation of the silica nanoparticles?

- Table 2. How have the temperature intervals been chosen?

  1. Section 3.3

- Lines 221-223. This result is in good agreement with the literature [26] demonstrating that the silica shell deposition was favored by the positively charged layer of gentamicin coating on the core particle surface. It is not clear which is the core particle surface. Is gentamicin deposited on silica nanoparticles or is the silica shell deposited on the gentamicin particles?

- Figure 3. The images must be on the same scale if the authors want to use them for to compare G0 to G2.

- Figure 4. As in figure 3, the image of G0 cannot be compared to the ones of G1 and G2.

- Figure 5.  Scale bars should be added on the pictures or in the legend.

- Lines 245-246. For G1, I don’t think particle aggregation is really obvious. Could the authors use another picture where it would be more visible?

- Line 248. The authors should try to be more specific on the groups that interact between the particles. Could those interactions be observed in FTIR?

- Lines 248-253 should be rephrased and explained more explicitly. According to the authors, addition of GS leads to pores filling, but also increases the pores (it is not specified whether in terms of quantity or in terms of pore size) and contributes to the improved intermolecular interactions inside the pore voids. The relation and causality between these events is not clear.

  1. Section 3.5

- Lines 262-264. In the case of testing the activity of the silica-gentamicin nanohybrid 262 (sample G2) on the Staplylococcus aureus and Escherichia coli strains, using the disc diffusion method, the studied material showed moderate activity

 Lines 266-268. Compared with the disc diffusion method, it can be observed that the silica-gentamicin nanohybrid (sample G2) possessed good antimicrobial activity against all the strains when the spot inoculation method was used.

The authors must explain how changing the testing method resulted in sample G2 turning from having a moderate activity to having a good activity. Which method should be trusted? What are the differences between the two methods?

- Additionally, how many times were the experiments performed? Are the inhibition zones average values?

- These strains are frequently used for antimicrobial testing. The authors must add data from the literature and compare their results to the ones they have obtained.

- Pure GS should also be presented as a comparison. If the results of GS at 24h are lower than the ones obtained with the GS-loaded NPs, an evolution of the inhibition with time should be included.

  1.  

There are a range of physicochemical characterizations that must be added to increase the scientific value and soundness of the present work. Solid-state NMR is a much better method to confirm GS incorporation compared to FTIR. Especially since GS seems to have been introduced in similar quantities compared to the silane precursors. Alternatively, EDS could confirm the distribution of GS on the particles. BET should be used to determine the porosity of the NPs and confirm that adding GS increases or decreases the porosity. XRD studies should be performed to observe the state of the drug in the particles (amorphous/crystalline). The drug loading must be calculated. The authors know how much GS has been used in the synthesis but the amount that is indeed incorporated in or on the surface of the particles is not calculated. Probably there are some losses in the synthesis media. The size of the particles must be measured by another technique except for microscopy, DLS for example.

The authors must include a scheme where the chemical formula of the silane precursor and the chemical formula of the final particles is given. Additionally, GS structure must be added somewhere in the manuscript.

10.

- There is no section 3.4.

- There are two figures 5.

  1. Finally English should be polished and there are also some typos. For example, but the list is not exhaustive: line 36 have and not has, line 41 because can, line 42 which allowing, line 58 the encapsulated of gentamicin, line 66 also, was demonstrated, line 85 the microwave approach proved is extremely fast, line 89 the successfully adsorption, line 96 has, line 103 to realize those materials, line 123 ), line 139 homogeneous and opalescence, line 172, line 232 can be understand, etc…

Author Response

In the present article, Purcar et al. report the synthesis, characterization and antimicrobial activity of gentamicin-loaded silica nanoparticles. The manuscript is concise, the microscopy studies are impressive, however the work lacks in terms of physicochemical characterizations.

  1. Abstract

- line 20: continuous microwave process. In general a continuous process refers to a process where there is a continuous stream of reactants and products in contrast to a batch process. The authors should use a different term.

Response

- Title of the manuscript was changed: “PREPARATION AND CHARACTERIZATION OF SILICA NANOPARTICLES AND OF SILICA-GENTAMICIN NANOSTRUCTURED SOLUTION OBTAINED BY MICROWAVE-ASSISTED SYNTHESIS

- “continuous microwave process” was replaced with “microwave-assisted synthesis”

  1. Introduction

- lines 87-89. The phrase “controlled power quickly to the reaction” should be rewritten. The phrase “uniform heating by mechanisms” is incomplete.

- lines 92-94. If the authors want to include this reference, they should elaborate slightly more on it.

Response

  • The phrase “controlled power quickly to the reaction” was rewritten:

“Microwave-assisted technique offer many benefits, such as: rapid volumetric heating, high reaction rates, spontaneous nucleation events, reproducibility, and size and shape control by tuning reaction parameters.”

  • Regarding this reference, more information’s were added: “They related that the increasing the microwave power from 100 to 450 W, lead at the alteration of the long-range ordering of the mesoporous structure of silica nanoparticles. The highest adsorption rate of 98.3% was obtained for mesoporous silica nanoparticles at 450 W, after 7 hours.”

  1. Section 2.2

- Quantities of TEOS, OTES and NH4OH, which I guess is the catalyst in the sol-gel process performed by the authors, should be expressed in moles (or mmoles).

- How can the authors explain the formation and isolation of nanoparticles? In general, tedious processes are implemented to obtain nanoparticles with a narrow size distribution, while in this manuscript is seems like the authors just added the silane precursors and particles are formed.

- Were the particles washed somehow to removed unreacted silane precursors?

- Lines 137-138: is gentamicin sulfate a liquid (in general salts are solids)? If it is a solid, authors should specify the solvent where GS was dissolved and the concentration of the solution. Why were these quantities used? Did the authors try other ones?

- Table 1. The theoretical content of gentamicin in the particles should be calculated and added to the table. The experimental content should be measured and added as well.

Response

  • Quantities of TEOS, OTES and NH4OH, were expressed in moles.
  • In this paper was shown that under microwave radiation the silica nanoparticles and silica-gentamicin nanostructured solution were obtained in short reaction time (40 minutes), compared with conventional methods.
  • These results indicate that the rapid and selective heating by microwave irradiation (temperature of 50 °C, microwave power of 200 W) is effective to make silica nanoparticles and silica-gentamicin nanostructured solution without the use of expensive agents.
  • The aim of this paper was to obtain silica-gentamicin solution that can be used as biochemical agent for the prevention and treatment of infections due to it effective biocidal activity and environmentally friendly performance.
  • Gentamicin sulfate was used as a liquid (solubilized in water, 3%). Other synthesis were not done at this moment in Institutes and Universities from Romania, because some of them are partially or totally closed due to the COVID-19 virus. Furthermore, other investigations requested by other researchers are not allowed to not spread the virus.

  1. Section 2.3

- If the authors have used a microwave probe, it should be specified somewhere.

- How many scans where taken for the IR spectra? The authors should also add how the spectra recorded (i.e., KBr tablets, nujol, something else).

 Response

  • Microwave probe wasn’t used in this research.
  • The information about FTIR analysis was added: “The structure’s modifications of the obtained materials were carried out on a Jasco FT-IR 6300 instrument (JASCO Int. Co., Ltd., Tokyo, Japan), with an integrating sphere, detector wide MCT (Pike Technologies Inc.). The samples were ground with KBr powder and pressed to form a disc for FTIR scanning. Data were collected in cooled liquid nitrogen, in a wavenumber range of 500–5000 cm1 (160 scans at a resolution of 4 cm1).”

  1. Section 3.1

- IR spectra are typically recorded up to 4000 cm-1 and interesting absorption bands can be observed in that region, why do the authors stop at 3200 cm-1?

- The peaks at 1525 and 1630 cm-1 are too weak to be used for characterization. Peak at 1525 cm-1 is not even indicated in the spectrum. A lot of peaks of similar intensity are observed in that region. Amine absorption should be observed around 3400-3600 cm-1 and, often, a different pattern is observed for primary and secondary amine groups. Additionally, to the best of my knowledge, gentamicin does not bear any aromatic rings. In any case, a structure of gentamicin must be added.

- In the 2850-2950 region, attributed to C-H stretching modes. Similar absorption bands should be observed in the GS spectrum, but there aren’t any. Why? Why is the intensity of the absorption band of G1 so much more intense compared to G0?

- According to the authors, the peak at 1620 cm-1 represents both loaded GS and adsorbed water. This peak is much too small to be used for such an assertion. Water in general is not observed in this region of the IR spectrum.

- To which specifical chemical bonds of nanosilica is the peak at 1460 cm-1 attributed to? Why is a similar peak observed on the spectrum of GS?

- To which chemical group is the band at 960 cm-1 attributed to? This peak is present in all three spectra.

- The small peak at 667 cm-1 could indeed be attributed to gentamicin and indicated the successful presence of GS in the silica particles, but the authors have to specify to which chemical group it belongs if they want to use it for their analysis.

- Finally, the authors concentrate on very weak peaks, while they do not comment at all on more intense peaks such as the peak around 1700 cm-1 in G1, the peak at ca. 1108 cm-1 in all spectra. These peaks must absolutely be attributed. Additionally, the peaks at 739 and 610 cm-1 should also be discussed.

 Response

  • Data were collected in cooled liquid nitrogen, in a wavenumber range of 500–5000 cm1 (160 scans at a resolution of 4 cm1)
  • New FTIR spectra were added.
  • Sample G2 was noted as G1, and sample G1 was removed.

  1. Section 3.2

- The TGA analysis of GS should be added in Figure 2.

- Why is the degradation of GS observed in G1 and not in G2, which, according to the authors, contains more GS?

- Why is the mass loss more important in G1 compared to G0? G1 should have a higher organic content since it also contains GS.

- Is the residual mass consistent with the amount of OTES that is used for the preparation of the silica nanoparticles?

- Table 2. How have the temperature intervals been chosen?

Response

New information’s about TGA analysis were added. TGA of gentamicin sulfate (as powder) was added.

In Table 2, the temperature interval was added.

  1. Section 3.3

- Lines 221-223. This result is in good agreement with the literature [26] demonstrating that the silica shell deposition was favored by the positively charged layer of gentamicin coating on the core particle surface. It is not clear which is the core particle surface. Is gentamicin deposited on silica nanoparticles or is the silica shell deposited on the gentamicin particles?

- Figure 3. The images must be on the same scale if the authors want to use them for to compare G0 to G2.

- Figure 4. As in figure 3, the image of G0 cannot be compared to the ones of G1 and G2.

- Figure 5.  Scale bars should be added on the pictures or in the legend.

- Lines 245-246. For G1, I don’t think particle aggregation is really obvious. Could the authors use another picture where it would be more visible?

- Line 248. The authors should try to be more specific on the groups that interact between the particles. Could those interactions be observed in FTIR?

- Lines 248-253 should be rephrased and explained more explicitly. According to the authors, addition of GS leads to pores filling, but also increases the pores (it is not specified whether in terms of quantity or in terms of pore size) and contributes to the improved intermolecular interactions inside the pore voids. The relation and causality between these events is not clear.

Response

In this study, silica nanoparticles were loaded with gentamicin in order to obtain silica-gentamicin nanostructured solution.

Figures 3, 4 and 5 were replaced.

New information’s were added: “Also, it can be observed agglomerations of particles, possible due the interaction and interconnection between gentamicin and silica nanoparticles (Si-O-Si network with methyl- or ethyl-organic groups) [37]. The interconnection (presence of silanol group at the surface of silica nanoparticles linked with the –COOH or –NHgroup of gentamicin) could be a surface phenomenon via weak interactions of van der Waals force or hydrogen bonding [38].”

- Section 3.3. was improved with new comments: “This can lead to the filling and increasing of pores  and dense microstructure after adding of gentamicin. Ballarre et al. [37] reported that the agglomerations of particles can promoted the increase of unevenness. Moreover, the agglomeration leads to particle sedimentation and provides instability. The adding of the gentamicin molecules in the obtained material led to the increasing of pores, possible due to increase of amount of gentamicinAlso, the microstructures of silica-gentamicin nanohybrids become dense due to the insertion of gentamicin in the mesopores as well as due to its adherence on the glass surbstrates [37, 38]. It was demonstrated that the adsorption of gentamicin molecules into mesoporous channels are based on the adsorptive properties of silanol groups while surface area of silica-gentamicin nano-solution are correlated with the amount of gentamicin adding [39].”

  1. Section 3.5

- Lines 262-264. In the case of testing the activity of the silica-gentamicin nanohybrid 262 (sample G2) on the Staplylococcus aureus and Escherichia coli strains, using the disc diffusion method, the studied material showed moderate activity

 Lines 266-268. Compared with the disc diffusion method, it can be observed that the silica-gentamicin nanohybrid (sample G2) possessed good antimicrobial activity against all the strains when the spot inoculation method was used.

The authors must explain how changing the testing method resulted in sample G2 turning from having a moderate activity to having a good activity. Which method should be trusted? What are the differences between the two methods?

- Additionally, how many times were the experiments performed? Are the inhibition zones average values?

- These strains are frequently used for antimicrobial testing. The authors must add data from the literature and compare their results to the ones they have obtained.

- Pure GS should also be presented as a comparison. If the results of GS at 24h are lower than the ones obtained with the GS-loaded NPs, an evolution of the inhibition with time should be included.

Response

Section 3.3. was improved with new comments.

“Also, this fact can be due to the aforementioned release profile of gentamicin from the silica-gentamicin nanostructured solution. Corrêa et al. [14] studied the antibacterial activity of silica-encapsulated gentamicin. They observed that the encapsulation method appeared to affect the activity of the S. aureus, in which the inhibition diameter increases from 14.93 mm (not-encapsulated) to 21.80 mm (1000 mg encapsulated gentamicin), possible due to the surface and textural characteristics of the obtained material.”

We don't want to make PLAGIARISM to show how perfect are the obtained analyzes or results. We made a synthesis different from what exists in the literature, in other working conditions. I have fixed some errors that have inevitably and unintentionally appeared in the making of this manuscript. Even if some researchers might help us, they should be included as co-authors. So, it means that a new manuscript will be made.

  1.  There are a range of physicochemical characterizations that must be added to increase the scientific value and soundness of the present work. Solid-state NMR is a much better method to confirm GS incorporation compared to FTIR. Especially since GS seems to have been introduced in similar quantities compared to the silane precursors. Alternatively, EDS could confirm the distribution of GS on the particles. BET should be used to determine the porosity of the NPs and confirm that adding GS increases or decreases the porosity. XRD studies should be performed to observe the state of the drug in the particles (amorphous/crystalline). The drug loading must be calculated. The authors know how much GS has been used in the synthesis but the amount that is indeed incorporated in or on the surface of the particles is not calculated. Probably there are some losses in the synthesis media. The size of the particles must be measured by another technique except for microscopy, DLS for example.

Response

Thank you for your special attention to this manuscript and for your suggestions. We appreciate all your comments.

Other analysis cannot be done now in Institutes and Universities from Romania, because some of them are partially or totally closed due to the COVID-19 virus. Furthermore, other investigations requested by other researchers are not allowed to not spread the virus. Please understand our views and support us in taking a favorable decision for this manuscript.

The authors must include a scheme where the chemical formula of the silane precursor and the chemical formula of the final particles is given. Additionally, GS structure must be added somewhere in the manuscript.

Response

The scheme was included (where the chemical formula of the silane precursor and the chemical formula of the final particles are given).

10.

- There is no section 3.4.

- There are two figures 5.

Response

The section and Figures were corrected.

  1. Finally English should be polished and there are also some typos. For example, but the list is not exhaustive: line 36 have and not has, line 41 because can, line 42 which allowing, line 58 the encapsulated of gentamicin, line 66 also, was demonstrated, line 85 the microwave approach proved is extremely fast, line 89 the successfully adsorption, line 96 has, line 103 to realize those materials, line 123 ), line 139 homogeneous and opalescence, line 172, line 232 can be understand, etc…

Response

The manuscript was improved and the grammar or English mistakes were corrected.

Round 2

Reviewer 1 Report

the authors have made a decent effort to address the shortcomings of the paper and while it could still be improved (still no positive control, and the comparison of the microwave synthesis are based solely on literature and not on their own data with only a single parameter presented) I believe it is suitable for publication

Author Response

Thank you for the positive comments.

New information’s were added:

  • page 3: “In our previously studies, we showed that the silica nanoparticles can be successful obtained through microwave-assisted synthesis (maximum power to 200 W, temperature of 50 °C), using silane precursors at different molar ratios (5:1 and 10:1, respectively). The results revealed that the obtained silica nanoparticles present different average particle sizes, ranging from 138.5 nm to 697.5 nm. The solvent, silica precursors, catalyst, and microwave irradiation time were proper selected to obtain modified silica nanoparticles with hydrophobic property [30].”

Reviewer 2 Report

The revision certainly improves the value of the work, introduces many essential details and convinced me of the correct work hypothesis.
I still disagree with the conclusion: "This silica-gentamicin solution obtained by microwave-assisted synthesis can be used as biochemical agent for the prevention and treatment of infections due to it effective biocidal activity and environmentally friendly performance." because such properties have not been demonstrated in the wider application context. These are prospects at best. Nevertheless, I believe that the work in its current form is complete and brings new, valuable knowledge, therefore I recommend publishing it. 

Author Response

Thank you for your suggestion and the positive comments.

The conclusion: "This silica-gentamicin solution obtained by microwave-assisted synthesis can be used as biochemical agent for the prevention and treatment of infections due to it effective biocidal activity and environmentally friendly performance." was replaced with:

  • page 13: "The silica-gentamicin solution obtained by microwave-assisted synthesis can be used as biochemical agent for the prevention and treatment of microorganisms which are deposited on different surfaces (e.g. glass, plastic, ceramic)."

Reviewer 3 Report

The new version of this manuscript has been greatly improved by the authors, and most of the concerns I expressed in my previous report have been addressed.

I still have some remarks regarding the IR characterisation.

Lines 231-233: In spectrum of gentamicin sulfate (GS), two peaks at 1530 and ~1630 cm-1 were detected, representing the bending vibrations of N–H (vibrational bending of primary aromatic amines).

I must insist that neither gentamicin sulfate nor the silica nanoparticles bear any aromatic amines. Could the authors possibly mean cyclic amines or amine groups bonded to an (aliphatic) ring?

I would like the authors to comment on peak at 1122 cm-1 in the IR spectrum of G1 which is the most prominent peak of the spectrum.

There are still some typos and English mistakes and I would advise a good and thorough review of the manuscript for those (eg lines 282, 307, 314, 317, 324, 331, 359,...).

Author Response

Thank you for your suggestion and the positive comments.

The new information’s were added:

  • page 6, section 3.1.:

“In spectrum of gentamicin sulfate (GS), two peaks at 1530 and ~1630 cm-1 were detected and it can be assigned to cyclic amines or amine groups bonded to an aliphatic ring.”

“The characteristic peaks at 1122 cm-1 and 1045 cm-1 were assigned to asymmetric Si–O–Si stretching vibrations [18, 34].”

The manuscript was improved and the grammar or English mistakes were corrected.
